# The seasonal and interannual variabilities of the barrier layer thickness in the tropical Indian Ocean

Xu Yuan*, Zhongbo Su

Faculty of Geo-Information Science and Earth Observation (ITC), University of Twente, the Netherlands

*Correspondence to:* Xu Yuan (x.yuan@utwente.nl)

**Abstract.** The seasonal and interannual variations of the barrier layer thickness (BLT) in the Tropical Indian Ocean (TIO) is investigated in this study using the Simple Ocean Data Assimilation (SODA) version 3 reanalysis dataset from 1980 to 2015. Seasonally, BLT anomalies in the western TIO (55°E-75°E, 5°N -12°S) are negatively correlated to SSS anomalies due to the change of the freshwater in boreal winter, spring, and autumn. In boreal winter, thermocline anomalies induced by the upwelling due to the local wind change positively correlates with BLT anomalies in the western TIO. In contrast, BLT anomalies are not well correlated with both SSS and thermocline anomalies in the eastern TIO (85°E-100°E, 5°N -12°S) in all four seasons. Prominent BLT and thermocline anomalies in the western TIO could be observed in both positive and negative Indian Ocean Dipole (IOD) years, while thicker BLT and deeper thermocline in the eastern TIO are prominent only in the positive IOD years. On the other hand, BLT in the western TIO presents a remarkable seasonal phase locking feature during the El Niño years. Thicker BLT in the western TIO is due to deepening thermocline induced by the westward Rossby wave during the developing and mature phases of El Niño, while thickening BLT opposing to the weakened thermocline change, becomes more significant due to the decreasing SSS induced by the freshwater during the decaying phase of El Niño.

## 1 Introduction

The upper-ocean variability, particularly the formation and variability of the mixed layer, can provide a broad perspective to better understand the air-sea interaction and the impacts of climate events on the marine ecosystem. The Tropic Indian Ocean (TIO) with a shallower thermocline in the west (Yokoi et al., 2012, 2008; Yu et al., 2005) and stronger interannual variation in the east (Li et al., 2003; Saji et al., 1999) comparing to the tropical Pacific and the Atlantic Ocean, provides a unique study region to investigate the variability of upper ocean.

Traditionally, the mixed layer depth was defined by the temperature (de Boyer Montégut et al., 2004). However, the mixed layer has recently been redefined by using the oceanic density (Kara et al., 2000; Mignot et al., 2007). The new definition results in novel terminology, the barrier layer. The barrier layer thickness (BLT) is defined as the depth from the mixed layer bottom to the top of the thermocline (Lukas and Lindstrom, 1991; Masson et al., 2002; Sprintall and Tomczak, 1992). BLT plays a key role in oceanic dynamics and air-sea interaction. For example, BLT isolates the density of the mixed layer from the cooling entrainment, helping to sustain the heat for the formation of the El Niño Southern Oscillation (ENSO) (Maes, 2002; Maes et al., 2006; Maes et al., 2005), as well as contributing to the formation of the different ENSO types (conventional ENSO and ENSO Modoki) (Singh et al., 2011). The spatial structure of BLT led by different Ekman drift (Thadathil et al., 2007;

Vinayachandran et al., 2002) is crucial for the formation of monsoon cyclones in the pre-monsoon season
(Masson et al., 2005; Neetu et al., 2012).
The variability of BLT is attributed to various mechanisms, such as heavy precipitation, oceanic currents, wind
stress, and oceanic waves (Bosc et al., 2009; Mignot et al., 2007). For instance, thicker BLT mainly locates in
the areas beneath the Intertropical Convergence Zone (ITCZ) due to abundant rainfall (Vialard and Delecluse,
1998) or regions with large river runoff (Pailler et al., 1999). The strong wind stress anomalies could also
contribute to thickening the BLT (Seo et al., 2009).
In the TIO, the features of BLT seasonal and interannual variation have been found very similar to that of Sea
surface salinity (SSS). Firstly, the southeastern Arabian Sea, the Bay of Bengal, and the southeastern TIO,
characterized by significant seasonal variability of SSS due to different hydrological processes, are also observed
with the strong BLT seasonal variations (Schott et al., 2009). Secondly, it is reported that the seasonal variability
of SSS in the TIO is mainly attributed to the freshwater (precipitation and runoff) and the horizontal advection
(Rao, 2003; Subrahmanyam et al., 2011; Zhang et al., 2016; Zhang and Du, 2012), whereas the BLT annual
variation could also be affected by the freshwater (Masson et al., 2002; Qu and Meyers, 2005). Moreover, the
Indian Ocean Dipole (IOD) and ENSO have been intensively reported to have impacts on the interannual
variability of SSS in the TIO (Grunseich et al., 2011; Rao and Sivakumar, 2003; Subrahmanyam et al., 2011;
Yuhong et al., 2013), while the IOD events can also partly explain the interannual variability of BLT in the
southeastern TIO (Qiu et al., 2012). In general, in the positive IOD year (*e.g*., 2006), the isothermal layer is lifted
by the upwelling Kelvin wave, and the mixed layer becomes shallower due to salinity decrease, which results in
a thinner barrier. In the negative IOD year (*e.g*., 2010), a thicker BLT is expected due to the extending of the
isothermal layer. Furthermore, the tight relationship between BLT and zonal SSS gradient was also reported in
the TIO at the sub-seasonal time scale. The zonal SSS gradient, led by the advection-driven freshwater, produces
a thicker BLT, which in turn, sustains a fresh and stable mixed layer depth (MLD) (Drushka et al., 2014).
However, existing studies on the interannual variability of BLT were mainly for specific years and lack of long-
term verification and the co-varying between the BLT and SSS variation is not always held in the TIO (Qiu et
al., 2012). Also, the relationship between BLT and the thermocline anomaly is less documented in the TIO. As a
result, further investigations of BLT seasonal and interannual variabilities and its relationship with SSS and
thermocline anomalies are still highly desired. The Simple Ocean Data Assimilation (SODA) version 3
reanalysis dataset from 1980 to 2015 could be adequate for such purpose.
The remainder of this paper is arranged as follows. After a description of the datasets and methods in section 2,
we compare the variability features of BLT obtained from observed and reanalysis datasets in the Indian Ocean
in section 3. Section 4 presents the seasonal variation of the BLT anomalies in the TIO. Its interannual variability
is shown in section 5. A summary and discussion are given in section 6.
**2 Data and Methods**
A series of monthly global gridded observation and reanalysis products were used to assess the variability of
BLT in the Indian Ocean. At 1° horizontal resolution, this includes the Argo profiles products provided by the
French Research Institute for Exploration of the Sea (Ifremer:
[http://www.ifremer.fr/cerweb/deboyer/mld/Subsurface_Barrier_Layer_Thickness.php](http://www.ifremer.fr/cerweb/deboyer/mld/Subsurface_Barrier_Layer_Thickness.php)) from 2005 to 2015. BLT
is calculated as the difference between $TTD_{DTm02}$ and $MLD_{DReqDT02}$.

$$BLT = TTD_{DTm02} - MLD_{DReqDT02}$$

where $TTD_{DTm02}$ is the top of thermocline depth defined as the depth at which the surface temperature cooled by
0.2 ℃, which is also referred to as the Isothermal Layer Depth (ILD) henceforth. $MLD_{DReqDT02}$ is the mixed
layer depth defined in oceanic density by assuming the density within the layer is 0.03 kg/m³ smaller than that of
the surface (de Boyer Montégut et al., 2007; Mignot et al., 2007).
At 0.5° horizontal resolution, The latest released version 3 SODA ocean reanalysis data (1980-2015) provided by
the Asia-pacific data-research center (APDRC: [http://apdrc.soest.hawaii.edu/datadoc/soda_3.3.1.php](http://apdrc.soest.hawaii.edu/datadoc/soda_3.3.1.php)) is
employed in this study. SODA version 3 has reduced systematic errors to the level that are adequate for the no-
model statistical objective analysis in the upper ocean and also has improved the accuracy of poleward
variability in the tropic (Carton et al., 2018). It has 26 vertical levels with a 15-m resolution near the sea surface.
We adopted the same Ifremer equation to calculate SODA BLT by using the mixed layer depth defined by both
the density and temperature.
We take the salinity and temperature in the first level (5m) as the SODA SSS and SST, respectively. The
thermocline depth is defined as the depth of 20 ℃ isotherm.
Monthly sea surface temperature (SST) obtained from Hadley Center Global Sea Ice and Sea Surface
Temperature (HadISST: [https://climatedataguide.ucar.edu/climate-data/sst-data-hadisst-v11](https://climatedataguide.ucar.edu/climate-data/sst-data-hadisst-v11)) for 1980-2015 on a
grid of 1°x1° is also used for the validation purpose and to calculate the Nino3.4 index (5°N-5°S,170°W-
120°W).
In all the datasets, we removed the annual cycle before proceeding with correlation. The simultaneous and lead-
lag correlations are evaluated in this study with the *t*-student significance test. The composition analysis is
employed in studying the interannual variability of BLT with the Monte-Carlo significance test. For each month,
the IOD/El Niño/La Nina years are randomly shuffled (10 000 times) and a mean *t*-student significance test t
statistic for the selected areas is calculated. The mean of the t statistic generated by the random simulations
exceeding that of the actual t value is determined and assessed at the 5% significance level. The positive and
negative IOD years are provided by the Bureau of Meteorology ([http://www.bom.gov.au/climate/iod/](http://www.bom.gov.au/climate/iod/)) and the El
Niño and La Nina years are obtained from Golden weather gate service ([https://ggweather.com/enso/oni.htm](https://ggweather.com/enso/oni.htm)).
Monthly mean fields are averaged over a three-sequential month for different seasons, *e.g.*, December-January-
February (DJF) for boreal winter, March-April-May (MAM) for boreal spring, June-July-August (JJA) for boreal
summer and September-October-November (SON) for boreal autumn. All the area-averaged parameters shown
in this study are weighted by the cosine of the latitude.

**3 BLT in the Indian Ocean**

BLT calculated by SODA version 3 reanalysis data is assessed against Argo float observation from 2005 to
2015. Figure 1 shows the distributions of the climatological BLT in the TIO for different seasons. BLT
climatology obtained from SODA presents a thinner bias in the Bay of Bengal in all four seasons comparing to
Argo BLT. This weakened BLT is probably because of lacking the runoff data in the Bay of Bengal (Carton et
al., 2018; Carton and Giese, 2008, 2006). SODA BLT fails to capture the BLT feature in the western TIO and
northwestern Arabian Sea where no BLT is expected (white areas with green line). However,  for the area of
interest, the BLT in SODA shows a coherent spatial pattern with the Argo BLT in the TIO (55°E-100°E, 5°N -
12°S). For instance, thicker BLT locates in the eastern TIO while thinner BLT locates in the western TIO. The
seasonal evolution of BLT in the eastern TIO obtained from SODA is consistent with that from Argo as well.
The area and intensity of BLT in the eastern TIO experience decreasing from boreal winter to spring and
increasing in both boreal summer and autumn.
To evaluate the seasonal and interannual variabilities of SODA BLT, the region-averaged BLT over two
separated boxes ( from 55 °E -80°E and from 85°E to 100°E, respectively) along with the band between 5°N and
12°S is shown in Figure 2 and Figure 3. In Figure 2, both SODA reanalysis and Argo capture the seasonality of
BLT, although the details are somewhat different. In the west sector (55°E -80°E), the thickest BLT is in boreal
winter while relatively thin BLT is in boreal spring. In contrast, in the eastern sector (85°E-100°E), the relatively
thick BLT occurs in boreal autumn while the thin BLT occurs in boreal spring and summer.
Due to the insufficient temperature-salinity observations, we only compare the interannual variability of the
SODA BLT with the Argo during 2005-2010. Two curves show good consistency in both west of 80°E and east
of 80°E (Figure 3). Respective correlations between SODA and observation for the west of 80°E and east of
80°E are 0.75 and 0.90, which are statistically significant at the 99.9 % confidence level.
Thus, comparisons between SODA and Argo BLT show the SODA capability in representing the seasonal and
interannual variability of the BLT in the TIO. In the next section, we will only use SODA reanalysis data to
investigate the seasonal and interannual variability of BLT in the TIO (55°E-100°E, 5°N -12°S) from 1980 to
131   2015.

The seasonal and interannual variations of MLD and ILD averaged over the west sector (55°E-80°E, 5°N -12°S)
and the east sector (85°E-100°E, 5°N -12°S) have also been calculated and presented in Figure 4 to investigate
the dominant driver for the BLT variability. However, it is hard to conclude either MLD or ILD as the main
dominator. In particular, both MLD and ILD display an annual cycle while BLT presents a semi-annual cycle in
the western sector. In the eastern sector, both MLD and ILD increase from March to August and decrease from
September to February, while BLT increases from March to November and decreases from December to
February. Thus, the impacts of MLD and ILD on the BLT is dependent on the seasons. On the other hand,  from
their interannual time series, there is no BLT (negative) in the years with deeper MLD, while prominent BLT
exists in the years with deeper ILD. We also calculated the correlation coefficients between BLT and MLD and
ILD, which are -0.07 and 0.47 in the west sector and -0.25 and 0.38 in the east sector. The interannual variation
of BLT is mainly related to the ILD variation in the TIO. To further study the seasonal and interannual variations
of BLT, we choose the variables in MLD, such as SST and SSS, and thermocline (prominent variations in the
deeper ocean).

**4 Seasonal variation**

To understand the seasonal variability of BLT in the TIO, Figure 5 presents the distributions of SSS and BLT during boreal winter, spring, summer, and autumn. The BLT distribution pattern is inversely correlated to the SSS distribution, where thick BLT is observed with fresh water in the eastern TIO, and vice versa in the western TIO. This is consistent with previous studies (Agarwal et al., 2012; Felton et al., 2014; Han and McCreary, 2001; Vinayachandran and Nanjundiah, 2009). Although both SSS and BLT have a similar distribution in the TIO, their seasonal variabilities do not co-vary with each other, especially near the equator. For example, saltwater in the western sector (55°E-80°E, 5°N -12°S) elongates to the east during boreal winter and spring and retreats during boreal summer and autumn, while the corresponding thin BLT does not vary accordingly. In contrast, there is significant seasonal variability of BLT in the eastern sector (85°E-100°E, 5°N -12°S), with its maxima occurred in boreal autumn. Besides, there is a weak seasonal variability of the east-west SSS gradient along the equator (5°N -12°S) while the zonal BLT gradient becomes more significant in boreal spring and strongest during boreal autumn. Thus, the seasonal variability of BLT in the TIO is not always co-varying with SSS.

To do a more detailed correlation analysis, we averaged data along the TIO. This area is adequate to demonstrate the difference in the seasonal variability between SSS and BLT. In addition, the well-known area of Seychelles Chicago Thermocline Ridge [SCTR, (60°E-80°E, 12°S-5°S)] (Manola et al., 2015; Yokoi et al., 2012, 2008) and the eastern IOD area [IODE, (90°E-110°E,10°S-EQ)] are also within the selected regions.

BLT anomalies were obtained from SODA version 3 reanalysis data averaged along the TIO (12°S –5°N) as a function of longitude vs. time. Figure 6 displays the in-phase correlations of SST and SSS anomalies with BLT anomalies respectively. At the seasonal time scale, although it has been proven that SST has a tight relationship with thermocline (Yokoi et al., 2012), there is no significant relationship between SST and BLT anomalies in the western TIO (55°E-75°E, 12°S-5°S). Instead, a short-term (less than two months) negative relationship between BLT and SST anomalies can be observed in the eastern TIO (85°E-100°E, 12°S-5°S) during boreal winter, with colder water connecting to thicker BLT and vice versa (Figure 6a). This SST-BLT relationship can also be found by correlating with SST anomalies obtained from the HadISST (Figure not shown), except their negative correlated area is smaller. On the contrary, a remarkable negative in-phase SSS-BLT (blue shaded) relationship, shows in the western TIO during boreal winter and spring with saltier (fresher) water corresponding to thinner (thicker) BLT (Figure 6b), while there is no significant relationship between SSS and BLT anomalies in the east.

To further understand the seasonal variability of BLT anomalies, we use the lead-lag crossing correlation for BLT anomalies in respect to SSS anomalies in January (JAN), April (APR), July (JUL) and October (OCT). The significant lead-lag relationship between SSS and BLT anomalies mainly locates in the western TIO (Figure 7), where is consistent with their in-phase relationship (Figure 6). SSS anomalies in boreal winter and autumn not only have a certain in-phase relationship with BLT anomalies but could also result in the change of the corresponding BLT anomalies at least two months earlier (Figure 7a,d). For example, in the western TIO, saltier water in October is presented with thinner BLT and can result in thinner BLT in November and December; saltier water in January associated with thinner BLT could lead to thinner BLT until May. This leading relationship of SSS anomalies on BLT anomalies is also found in April, but with a weaker negative BLT feedback from January and February (Figure 7b). In July, it is only found the BLT feedback in the western TIO,

implying that the spring-time BLT anomalies can have a negative impact on the summer-time SSS anomalies
(Figure 7c).
Figure 8 shows the lead-lag crossing correlation between BLT and the thermocline anomalies. Thermocline
anomalies have a positive correlation coefficient with BLT anomalies. Particularly, deeper thermocline in
October is along with thicker BLT in the central and the eastern TIO and has a positive leading effect on the
BLT anomalies in November and December. Although deeper thermocline in January is associated with thicker
BLT in the eastern TIO, there is no leading impact of thermocline anomalies on the BLT anomalies in the
following months. In contrast, the remarkable in-phase and leading relationships between thermocline and BLT
anomalies in January can be seen in the western TIO, which has since weakened in April. A weaker leading
effect of thermocline anomalies on BLT anomalies in April appears in the eastern TIO. In July, there is little
correlation between thermocline and BLT anomalies in the TIO.
According to the above analysis, we examined the corresponding atmospheric forcing in the western TIO and
eastern TIO, respectively. Figure 9 shows the seasonal evolution of the upper-ocean salinity, MLD, ILD,
thermocline, freshwater flux (Precipitation minus Evaporation, P-E), and the zonal component of the wind stress.
In the western TIO, freshwater flux freshens the upper-ocean water from October to April, which in turn,
thickens the BLT, consistent with the analysis in Figure 7. On the other hand, westerlies lead to Ekman pumping,
which in turn, results in the thinner thermocline (green line) to affect the BLT. In the eastern TIO, the seasonal
variation of BLT is more complex than that in the western TIO. Firstly, the seasonal evolution of SSS has a
semi-annual feature, while freshwater flux does not.  This may link to the Indonesian throughflow which brings
freshwater from the Pacific Ocean into the eastern TIO (Shinoda et al., 2012). Secondly, the thermocline
presents the opposite seasonal cycle comparing with that in the western TIO, although the zonal wind stress
displays a similar seasonal variation in both the western and eastern TIO. Last but not least, we also noticed that
the salinity in the deeper ocean varies similar to the thermocline in the eastern TIO, which is different in the
western TIO. Thus, the seasonal variation of BLT in the eastern TIO is not mainly driven by freshwater flux and
wind-driven upwelling. Felton et al. ( 2014) have suggested that the seasonal BLT variation in the eastern TIO
may be related to the sea level and ILD oscillation.
In summary, during boreal autumn, BLT anomalies in the eastern TIO are determined by the subsurface oceanic
process while SSS anomalies drive BLT anomalies in the western TIO with the negative in-phase and leading
impacts. During boreal winter, due to strong wind convergence induced by both the winter monsoon wind and
the southeasterlies (Yokoi et al., 2012), BLT anomalies are affected by both SSS and thermocline anomalies in
the western TIO. SSS anomalies have a negative influence on BLT anomalies while thermocline anomalies have
a positive impact. This negative SSS-BLT relationship sustains until boreal spring with weaker negative
feedback of the BLT anomalies on the surface. A relatively weaker thermocline-BLT relationship is observed in
the eastern TIO. SSS in boreal summer is affected by the spring-time BLT in the western TIO without the BLT-
thermocline relationship.
**5 Interannual Variation**
IOD, as the zonal SST gradients along the equatorial TIO, is a crucial climate mode on the interannual time scale
(Schott et al., 2009). It corresponds well with local precipitation and wind change and has impacts on the SSS
(Saji and Yamagata, 2003a). IOD events mostly develop and mature within the boreal autumn and decay in
boreal winter (Saji et al., 1999). The intensity of IOD could be defined by the Dipole Mode Index (DMI), which
is the difference between SST anomalies in the region of (10°S –10°N, 50°E-70°E) and (10°S –EQ, 90°E-110°E)
(Saji et al., 1999). Accordingly, we composited the monthly SSS, BLT and thermocline anomalies for positive
IOD (pIOD) events and negative IOD (nIOD) evens based on DMI. The corresponding years are listed in Table
1. Figure 10 presents the composited seasonal variations for our current dataset during the period of 1980-2015.
The Monte-Carlo procedure has been used to evaluate the significance of the composite variations (green shaded
areas). If a signal exceed the green shaded areas, it is assessed significant at the 95% significance level. In the
eastern TIO (Figure 10a,c,e), there is no significant signal of SSS during the IOD events while thermocline
accompanying BLT displays the prominent seasonal phase locking feature. Both thin BLT and shallow
thermocline appear during the mature and decaying phases of the positive IOD events due to the reduced
precipitation and the strong upwelling (Thompson et al., 2006), which in turn, contributes to intensifying the
positive IOD events coupled with SST (Deshpande et al., 2014). Thicker BLT can be found in the mature phase
of the negative IOD events along with a deeper thermocline. In the western TIO (Figure 10b,d,f), SSS, BLT, and
thermocline only respond well to the positive IOD events. BLT has been thickened due to deeper thermocline
and fresher water, providing favorable circumstances to sustain warmer water.
The interannual variability of the TIO is remotely affected by ENSO. A significant seasonal phase-locking
impact of ENSO on the TIO has been addressed in previous studies (Schott et al., 2009; Zhang and Yang, 2007).
Normally, there are three phases of ENSO, namely the developing phase of ENSO (boreal autumn), the mature
phase of ENSO (boreal winter) and the decaying phase of ENSO (boreal spring). We composited our variables
based on the ENSO events from Table 2. Figure 11 presents the composited results of the seasonal variation. In
the eastern TIO (Figure 11a,c,e), the thinner BLT mainly connects to the shallower thermocline during the
developing and mature phases of El Niño (Figure 11c, e), due to the anomalous easterlies along the equator
invoked by the adjusted Walker Circulation (Alexander et al., 2002; Kug and Kang, 2006). In the western TIO
(Figure 11b,d,f), the deepening thermocline, due to the westward downwelling Rossby wave and the anomalous
wind stress induced by El Niño (Kug and Kang, 2006; Xie et al., 2002), peaks in the El Niño developing phase.
A following weak peak of thermocline anomalies appears during the decaying phase of El Niño events. The
corresponding BLT has a similar semi-annual variation in the El Niño years, but with the intensified second peak
after the mature phase of El Niño attributed to the fresher surface water.
The seasonal phase locking of BLT is prominent in the eastern TIO mainly induced by the corresponding
thermocline in both the IOD and the ENSO years. In the western TIO, the variation of BLT is influenced by
thermocline during the developing and mature phases and affected by SSS during the decaying phase of the
positive IOD or El Niño events.
The relationship between BLT and El Niño could also be detected in the time series of BLT, SSS and
thermocline anomalies averaged over the western TIO during boreal winter and spring from 1980 to 2015
(Figure 12). During boreal winter (Figure 12a), deeper BLT and thermocline could be found in 1983, 1992,
1998, corresponding to the mature phase of El Niño. During spring (Figure 12b), deeper BLT and thermocline
could also be observed in the decaying phase of El Niño years, accompanying with fresher water. On the other
hand, the effect of IOD on the interannual variability of BLT could be observed in specific years as well, such as
1983,1998 and 2006.
Next, we calculate the lead-lag correlations of BLT, thermocline and SSS anomalies with the Nino3.4 index
(averaged SST anomalies in the area of (5°N -5°S, 170°W -120°W)) from 1980 to 2015. The BLT-El Niño
relationship experiences two phases (Figure 13a) linking to the subsurface and surface effects. In particular, the
correlation coefficients between the thermocline anomalies and the Nino3.4 index reach the noticeable values
during the mature period of El Niño (Figure 13b), and their correlation has a longitude-dependent time-delay,
which is consistent with the result of Xie et al., (2002). This deeper thermocline resulted by El Niño via the
westward downwelling Rossby wave affects the corresponding BLT anomalies, shown as one month later of the
remarkably positive correlation between BLT anomalies and the Nino3.4 index in Figure 13a. Then, the
correlation between the thermocline anomalies and El Niño becomes weaker during the decaying period of El
Niño. However, there is an enlarged correlation between BLT and ENSO, corresponding to the intensifying
second peak of BLT shown in Figure 10d. This enlarged pattern accompanies with the appearance of a negative
correlation between SSS and ENSO (Figure 13c). The negative SSS anomalies induced by El Niño via the
adjusting Walker circulation and the westward Rossby wave in the western TIO thicken the BLT anomalies
(Figure 13d,e).
In conclusion, according to the theory of Xie et al. (2002), there is warmer water developing in the eastern
tropical Pacific Ocean (El Niño), resulting in the anomalous easterlies and invoking the downwelling Rossby
wave along the equatorial TIO (Figure 13e). Thereby, thermocline has been deepened in the western TIO
associated with the thicker BLT. This thickening BLT hampers the upwelling process and benefits to sustain
warmer SST. On the other hand, there is an anomalous ascending branch of the Walker circulation adjusted
during the mature phase of El Niño. As a result, abundant precipitation forms over the TIO, impacting on SSS.
Consequently, fresher surface water helps to thicken BLT, which in turn, prolongs the warmer SST in the
western TIO.
**6 Summary**
The seasonal and interannual variability of BLT in the TIO was investigated mainly by using the SODA version 3
reanalysis dataset from 1980 to 2015. Although SODA differs in representing the no BLT status near the land mass
in the western TIO as shown in Argo, the SODA BLT displays the spatial feature in a good agreement with the
Argo BLT. Also, the seasonal and interannual variations of BLT in SODA is consistent with that in Argo. Despite
the biases in the spatial feature and variabilities of BLT, SODA is deemed to reproduce overall reasonably well
the main characteristics of the BLT in the TIO, and thus it has merits for further exploration of the long-term
seasonal and interannual variability of the BLT in the TIO.
The contributors to the seasonal variability of BLT is different between the eastern and western TIO. In the
eastern TIO, BLT is weakly affected by thermocline change, shown as the deeper thermocline leading to the
thicker BLT. This positive correlation between BLT and thermocline is prominent in boreal autumn. In the
western Indian Ocean, the factors affecting the BLT change with the season. During boreal autumn, SSS has a
remarkably negative correlation with the BLT. The saltier the water is, the thinner the BLT is. Both SSS and
thermocline anomalies have contributions to the BLT during boreal winter through the freshwater flux and the
winter monsoon wind-driven upwelling. The positive SSS anomalies shoal BLT while the positive thermocline
anomalies thicken BLT. During boreal spring, BLT anomalies are mainly driven by SSS. Meanwhile, there is a
weak BLT feedback on SSS anomalies, which is intensified in boreal summer.
In terms of the interannual variation, thicker BLT is distinct in the negative IOD and the La Nina years while
thinner BLT occurs in the positive IOD and the El Niño years. On the other hand, the prominent BLT shows
clear seasonal phase locking during the IOD and El Niño years. Particularly, in the eastern TIO BLT co-varies
with thermocline during the mature phase of both the IOD and El Niño events. Both SSS and thermocline
contribute to the change of BLT in the western TIO after the mature phase of the positive IOD events, and their
impacts on the BLT variation are enhanced in the El Niño years. In general, the warmer water in the tropical
Pacific Ocean deepens thermocline in the western TIO, resulting in thicker BLT. The correlation between
thermocline and El Niño becomes weaker during the decaying phase of El Niño, but the pattern of the correlation
between BLT and El Niño is enlarged attributed to the variation of SSS. Fresher water induced by the abundant
precipitation due to El Niño thickens the BLT after the mature phase of El Niño.

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

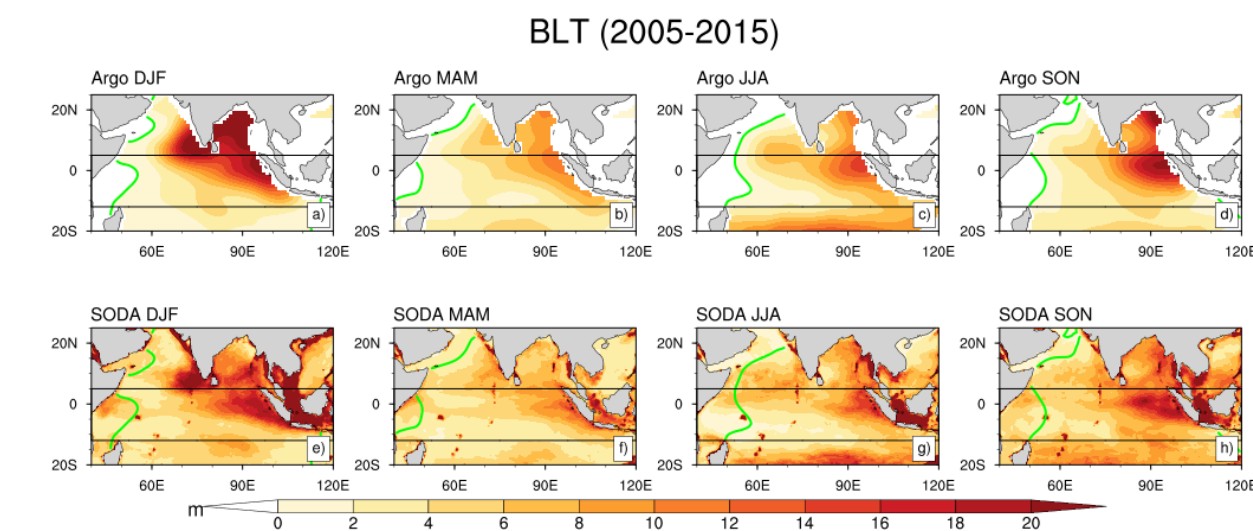



**Figure 1. Seasonal distributions of the BLT climatology obtained from Argo (a) and SODA (b) from 2005 to 2015 in**
**the Indian Ocean. Units: m. The thicker green line is the zero BLT line from Argo.**

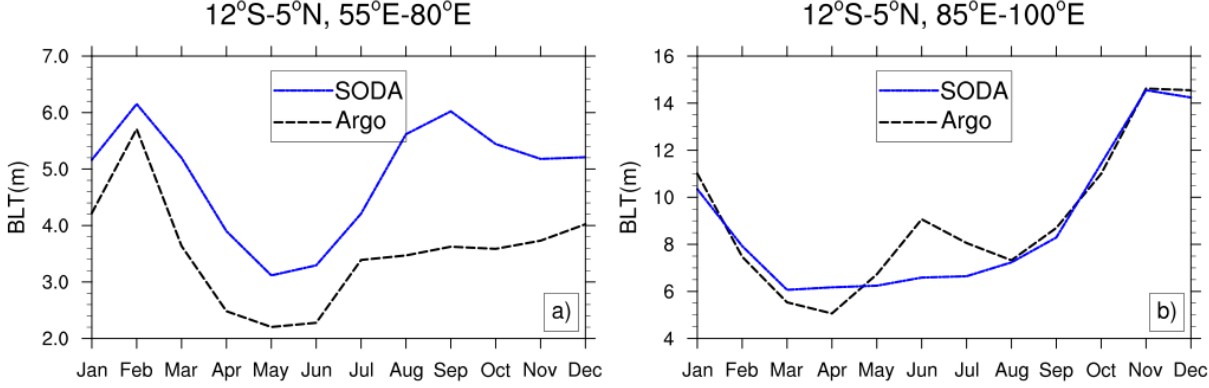


**Figure 2. Seasonal cycle of the region-averaged BLT for SODA and Argo: a) from 55°E to 80 °E and 12°S and 5°N,**
**and b) from 85°E to 100 °E and 12°S and 5°N.**

**Figure 3. Interannual time series of the region-averaged BLT for SODA and Argo: a) from 55°E to 80 °E and 12°S**
**and 5°N, and b) from 85°E to 100 °E and 12°S and 5°N.**

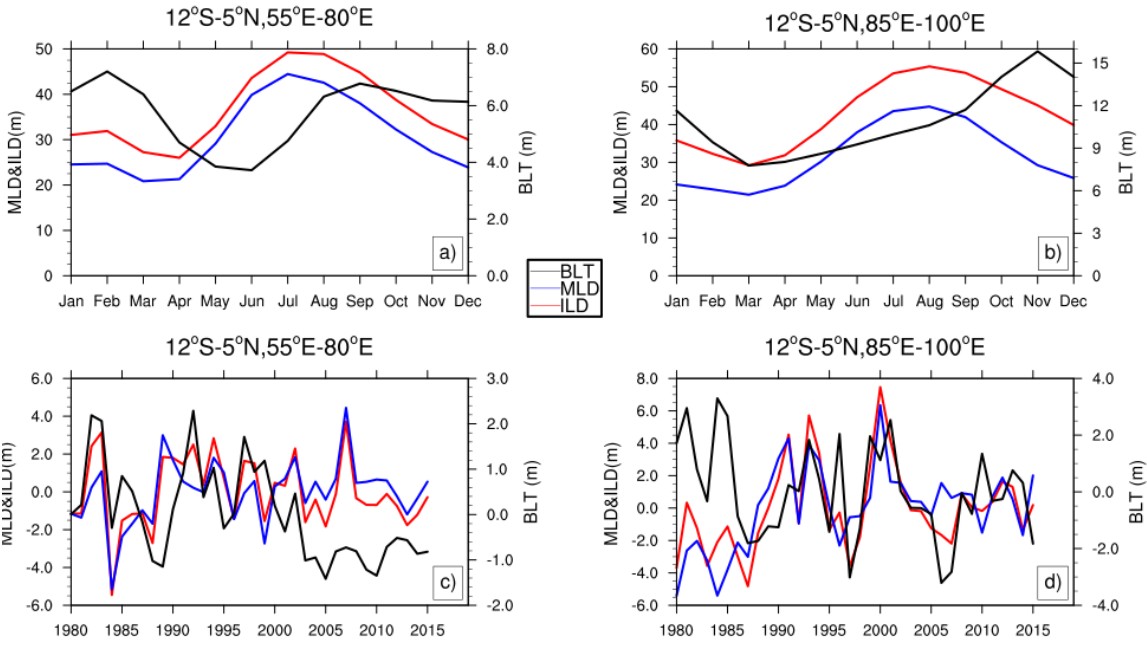


**Figure 4. The seasonal and interannual variations of BLT, MLD and ILD : a,c) from 55°E to 80 °E and 12°S and 5°N,**

**and b,d) from 85°E to 100 °E and 12°S and 5°N.**

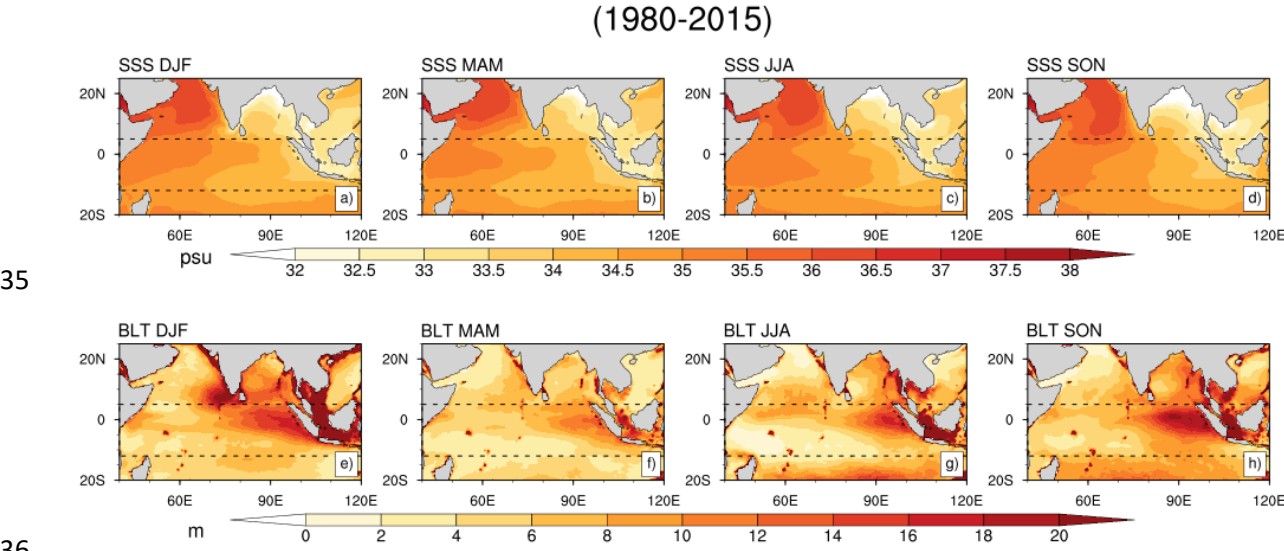



**Figure 5. The seasonal distributions of SSS (unit: psu; a-d) and BLT (unit: m; e-h) in the Indian Ocean from 1980 to**

**2015. The two dashed black lines represent the latitudes of 12°S and 5°N, respectively.**

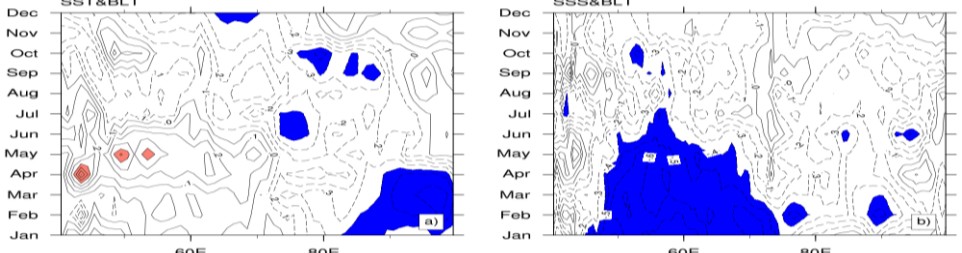


**Figure 6. Simultaneous correlations along the area of (12°S-5°N) for (a) SST and (b) SSS anomalies with respect to**
**BLT anomalies. Shaded areas exceed the 95% significance level, while the red and blue shaded areas represent the**
**areas with the positive and negative correlation coefficients, respectively.**

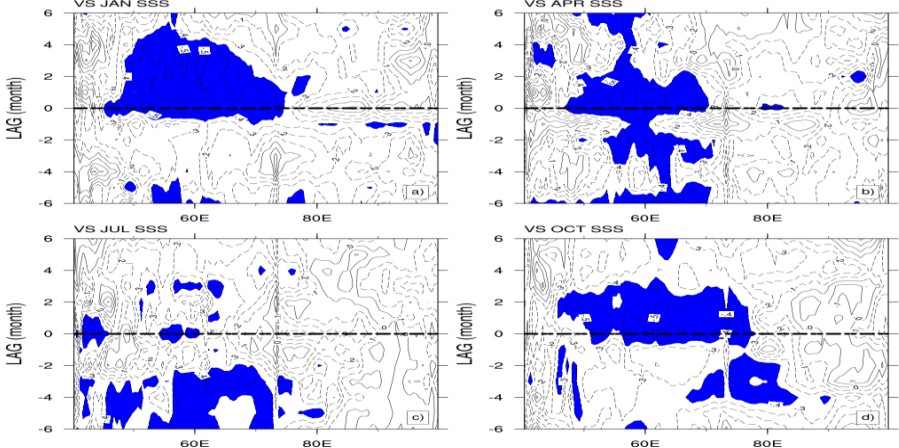


**Figure 7. Lead – lag crossing correlations between BLT and SSS anomalies for (a) January, (b) April, (c) July, and (f)**
**October along the area of (12°S-5°N) from 1980 to 2015. Shaded areas exceed the 95% significance level. Positive lag**
**means SSS leads BLT. Blue shaded areas represent the negative correlation. The thick black dashed line represents**
**the in-phase correlation.**

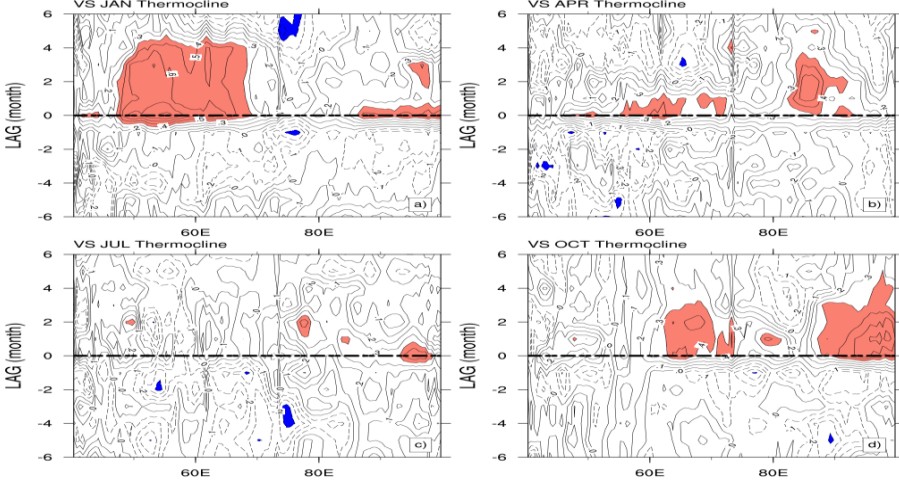


**Figure 8. Same as Figure 6 but for thermocline and BLT anomalies. Red (blue) shaded areas represent the positive**
**(negative) correlation. The thick black dashed line represents the in-phase correlation.**

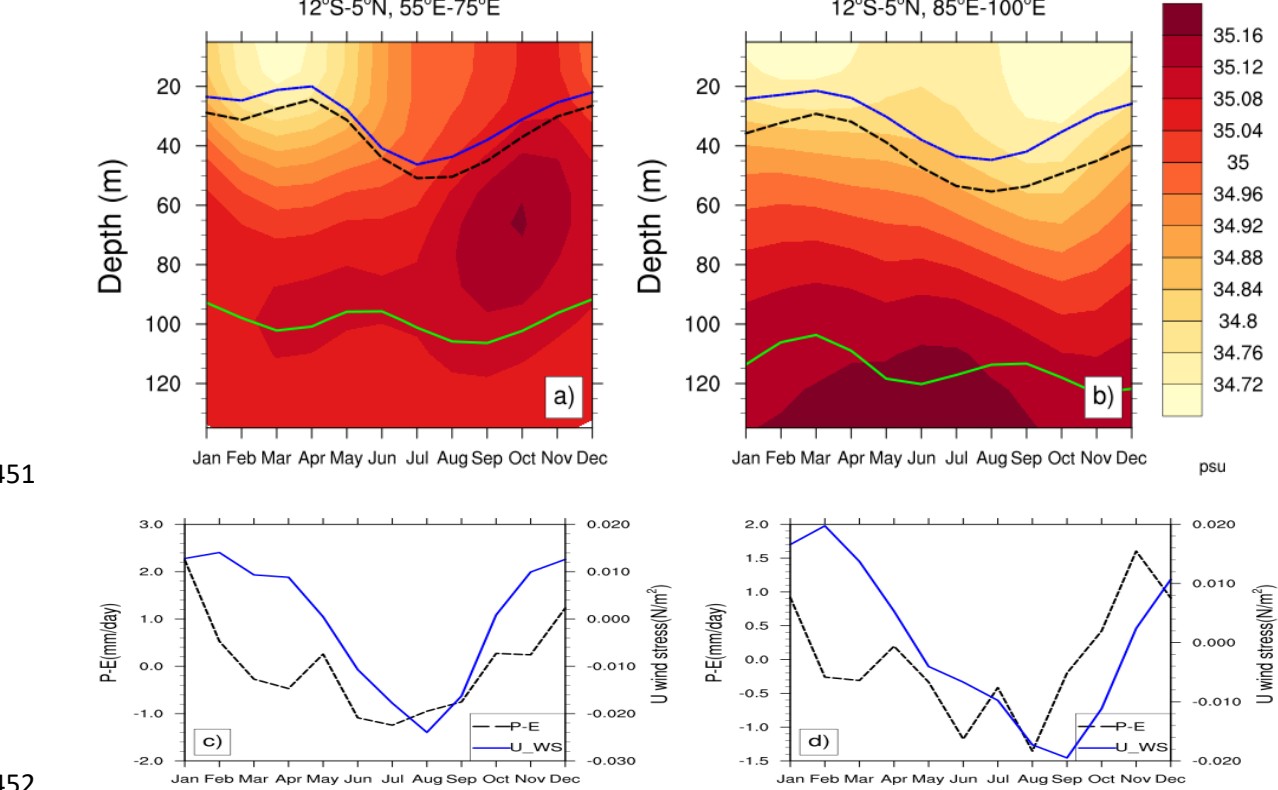



Figure 9. Seasonal variation in the (a,c)western TIO (12°S-5°N, 55°E-75°E) and (b,d) eastern TIO (12°S-5°N, 85°E-100°E). The top figures show the depth-time plots of the upper-ocean salinity (shaded), thermocline (green line), isothermal layer (black dashed line) and mixed layer (blue line). The bottom figures show the freshwater flux (P-E) and zonal component of the wind stress (U_WS) anomalies.

Table 1

List of positive IOD events and negative IOD events in our study period.

| pIOD years | 1982 | 1983 | 1994 | 1997 | 2006 | 2012 | 2015 |
|---|---|---|---|---|---|---|---|
| nIOD years | 1981 | 1989 | 1992 | 1996 | 1998 | 2010 | 2014 |

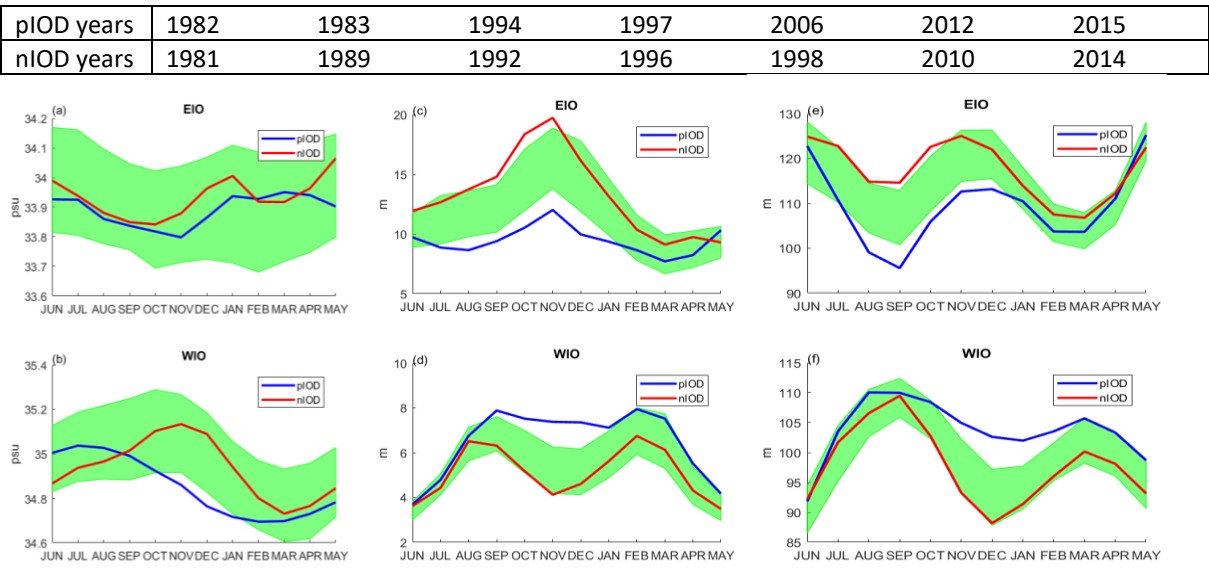

459

Figure 10. The compositing seasonal variations of SSS (a, b; unit: psu), BLT (c, d; unit: m) and thermocline (e, f; unit: m) in the IOD events during the period of 1980-2015 averaged by the areas of the eastern TIO (85°E-100°E, 12°S-5°N) and the western TIO (55°E-75°E,12°S-5°N), separately.The blue line represents compositing in the positive IOD events and the red one represents that in the negative IOD events and the green shaded area represents the 95% Monte-Carlo significance level.


**Table 2**
**List of El Niño events and La Nina events in our study period.**

| El Niño years | 1982 | 1987 | 1991 | 1997 | |
|---|---|---|---|---|---|
| La Nina years | 1988 | 1998 | 1999 | 2007 | 2010 |


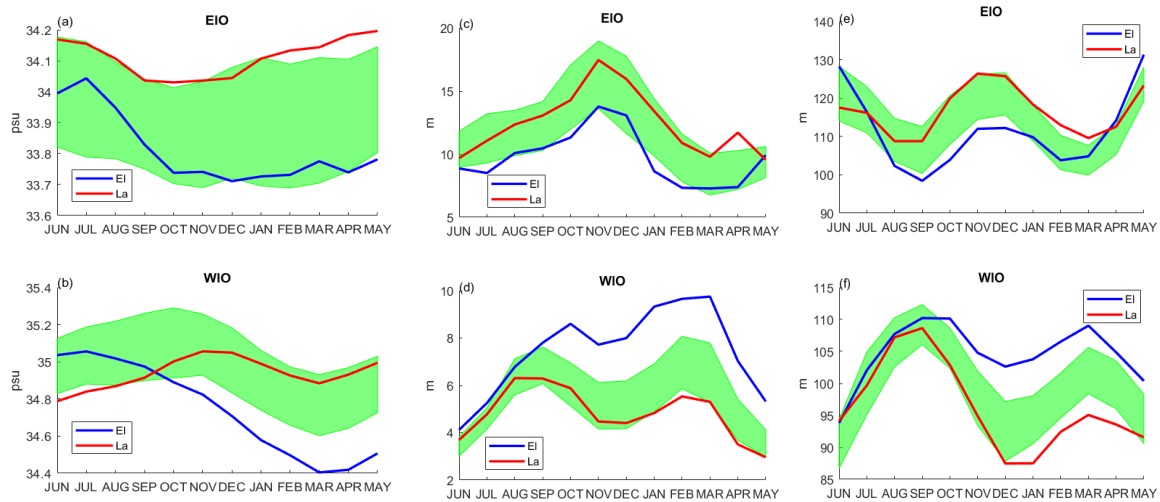


**Figure 11. Same as Figure 10 but compositing on the El Niño and La Nina years.**

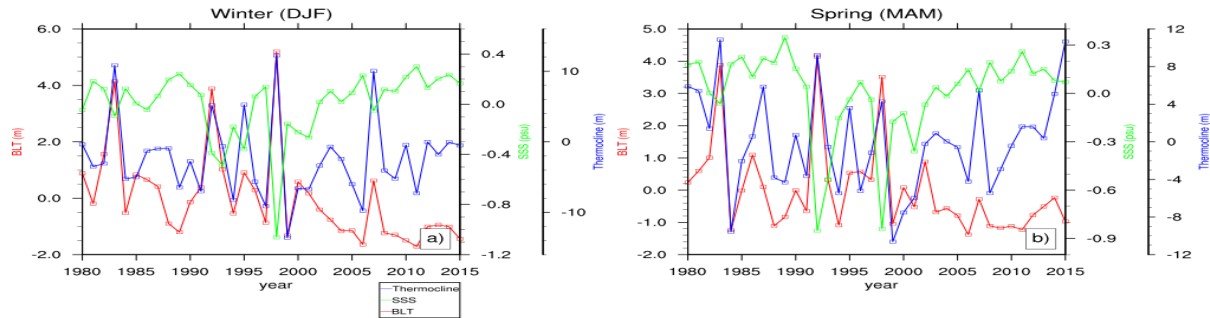


**Figure 12. Time series of BLT, SSS and thermocline anomalies averaged over the western TIO (12°S-5°N, 55°E-75°E)**
**during boreal winter (a) and spring (b) from 1980 to 2015. Red, green, and blue lines represent BLT, SSS and**
**thermocline, respectively.**

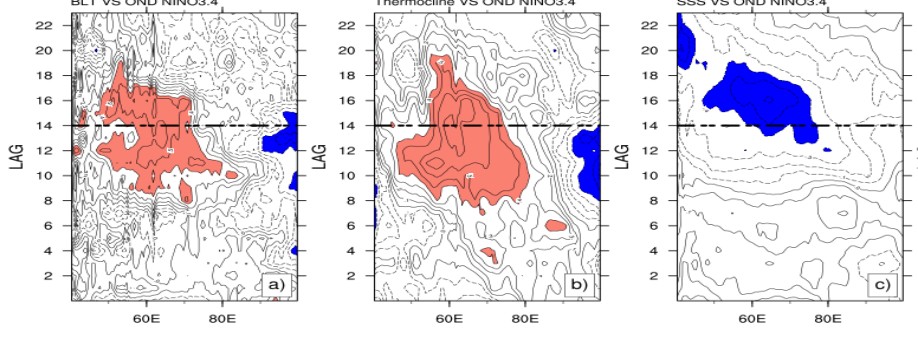


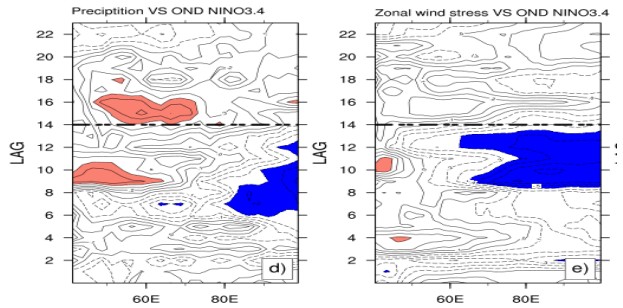


**Figure 13. Lagged correlations of (a) BLT, (b) thermocline, (c) SSS anomalies, (d) precipitation anomalies, and (e) zonal wind stress anomalies averaged in (12°S-5°N), with the Nino3.4 index as a function of longitude and calendar month (Shaded areas exceed 95% significance level; positive lagging correlations are shaded in red and negative ones are in blue; the thick black dashed line represents the start of the decaying phase of El Niño).**







