# Peer review of "Seasonal and interannual variabilities of the barrier layer"

_Ocean Science, 2019_

## Referee Comment (RC1) · Anonymous Referee #1 · 26 Jun 2019

This paper investigated the seasonal and interannual variability of BLT in the TIO by using SODA version 3 reanalysis dataset from 1980 to 2015. This research points out that the BLT shows a significant seasonal variation in the TIO, which is related to the variations of the sea surface salinity and the thermocline. The author also claimed both the Indian Ocean Dipole and ENSO events could impact the variation of BLT by affecting the thermocline in the TIO. Overall, the paper might be considered for publication once mandatory major changes are made. All my suggestions are listed below:

Major comments:

1. Line38-40: How about the definition: MLD in density with a variable threshold criterion (equivalent to a 0.2°C decrease)? Are there any differences by using the suggested new definition in calculating the MLD, and further for the BLT variation?

2. As the author explained, the BLT is defined by the difference of the MLD and ILD. What's the seasonal and inter-annual variation of these two aspects? Which one can mainly determine the BLT?

3. This paper mainly gave the seasonal and inter-annual variation of BLT in the TIO, but the mechanism for these was not explained enough. As the BLT is affected not only by the SST, SSS, thermocline, but also by the wind stress, rainfall or the fresh water input and even the net heat flux input. More work on mechanism analysis is encouraged.

4. The explanation of the impact of ENSO on the BLT variation is simply accorded to the theory of Xie et al. (2002). Did you really find the anomalous easterlies? The Walker Circulation is also needed to be verified.

5. Can you show the time series of the SSS, BLT and thermocline during the whole period of 1980-2015? Do the IOD or the ENSO event mainly contribute to the interannual variation?

Minor comments:

- 1. Line33-34: What variation?
- 2. Line 53-54: INTER-TROPICAL CONVERGENCE ZONE (ITCZ)?
- 3. Line 124: 2005-2015 => 2005-2015?

4. Section 3 might be too short. Author could explain more about figure 1 or move this section to the next section.

5. Line 213: The caption of figure 4: lead-leg?

6. Line 252: The lines could be plotted above the shaded area in figure 6.

СЗ

---

## Editor Comment (EC1) · A.J. George Nurser (Editor) · 20 Dec 2019

In this manuscript the authors examine the seasonal and inter-annual variation of the barrier layer thickness (BLT) in the Equatorial Indian Ocean, using output from the SODA v3 reanalysis. They consider the seasonal variation of BLT and its correlations with changes in SSS and thermocline depth. They then consider the inter-annual variability of BLT, and its correlation with the Indian Ocean Dipole (IOD) and the El Nino Southern Oscillation (ENSO).

Firstly, I am not entirely happy with the focus on the SODA reanalysis. It seems to give fields substantially different to ARGO, and I am suspicious of the positive BLT over the

whole domain given that substantial parts of the domain (esp to the NW) are areas of net FW loss where salinity should increase towards the surface and there should be no BL. I would like to see further validation of the SODA fields before I can accept this analysis. Also, the figures need to clearly differentiate areas of no BLT from no data; currently both are white, which is confusing.

It is interesting to see the seasonal variation of the BLT, but there is little discussion of the mechanisms driving it e.g. discussion of why does the SSS change should be linked to freshwater budget changes, while discussion of thermocline depth change should include e.g. details of changes in Ekman pumping. A couple of figures showing typical vertical profiles would also be useful.

The discussion of the interannual variability is a little sketchy, but is reasonable. In summary I would like substantial revision validating the data and emphasising more the mechanisms.

Detailed Comments

The English while readable, is still not great, and could do with reading by a native English speaker. E.g. "composting" actually means allowing vegetables to decay! I think you mean "compositing" or "composited"

p1, l28–36. Simplify to just stating that previous definitions in terms of temperature difference have been replaced by new definitions in terms of density difference; leave the details (0.2°, 0.03 kg/mˆ3 etc) to section 2.

p2, l23–24. Please explain why zonal SSS gradient is important. Is it to do with Ekman drift?

p3, l1–8. Please define much more carefully what your definitions of MLD are, and actually write them out as equations. Also, you should mention Kara et al. (2000), as the first paper to use a density criterion based on a temperature criterion: Kara, A. B., P. A. Rochford, and H. E. Hurlburt (2000), An optimal definition for ocean mixed layer depth,

J. Geophys. Res.-Oceans, 105(C7), 16,803–16,821, doi:10.1029/2000JC900072.

p3, l11 "reduced systematic errors to a level" What level?

p3 l30–35. BLT is zero in the western Indian Ocean in ARGO but not in SODA. SODA BLT see generally excessive. Please compare more fully with ARGO.

p4 l 3 and below. "winter' is confusing. Please use "boreal winter" or "northern winer" (at least the first few times).

p5 l 21–25. Please spent some time describing the IOD. A figure would help.

p6 l 1–10. Explain how developing and decaying El Nino are defined: it seems you are just considering El Nino and La Nina years.

p9, Fig. 1. Please differentiate between missing data and zero BLT.

p11, Fig. 6. Are green shaded areas 95% limits for all years? If so please state this clearly.

---

## Author Comment (AC1) · 20 Feb 2020

Dear Referee #1, We would like to thank you for your comments. Your valuable comments help us to improve the quality of this manuscript. In revising the paper, we have carefully considered all comments and suggestions. Major comments:

1. Line38-40: How about the definition: MLD in density with a variable threshold criterion (equivalent to a 0.2_C decrease)? Are there any differences by using the suggested new definition in calculating the MLD, and further for the BLT variation?

Many thanks for this comment. We have compared the calculated MLD using different

definitions with respect to its distribution and seasonal variation. For brevity, MLD defined by density with a fixed threshold criterion (0.03kg/m3) is termed as MLD_DR003 and MLD defined by density with a variable threshold criterion (equivalent to a 0.20C decrease) is short for MLD_D02. In terms of distribution, MLD_DR003 is shallower than MLD_D02 in the Indian Ocean, especially in the southwestern and eastern Indian Ocean (see Figure 1 below). The maximum difference between MLD calculated from those two definitions could reach 10 m difference. Although the seasonal change of MLD_DR003 and MLD_D02 is consistent with each other (Figure 2 below), MLD_DR003 is much shallower during boreal autumn, winter and early spring. Consequently, the seasonal variation of BLT may not be affected by the definition of MLD. But using deeper MLD_D02 to calculate may lead to negative BLT which is no physical meaning. Thus, we used the MLD_DR003 in this study.

2. As the author explained, the BLT is defined by the difference of the MLD and ILD. What's the seasonal and inter-annual variation of these two aspects? Which one can mainly determine the BLT?

Thank you for these comments. We have plotted the seasonal and inter-annual variations of MLD and ILD, respectively (Figure 4 in the revised manuscript). It is apparent that both the variation of MLD and ILD could have impacts on the BLT variation. But there is no robust correlation between the variations of either MLD or ILD and BLT. We conclude that the impacts of MLD and ILD on the BLT are season-dependent. For instance, the seasonal variations of MLD and ILD present an annual variation while the seasonal variation of BLT presents a semi-annual variation. The interannual variation of BLT is correlated better with the ILD change. To link to the atmospheric forcing, we mainly choose the SSS and SST which can be observed by satellites and reflect the characteristics of MLD and thermocline (which is beneath the ILD) to study the seasonal and interannual variations of BLT. The related analysis has been added in section 3 (Line 132-144).

"The seasonal and interannual variations of MLD and ILD averaged over the west sector (55°E-80°E, 5°N -12°S) and the east sector (85°E-100°E, 5°N -12°S) have also been calculated and presented in Figure 4 to investigate the dominant driver for the BLT variability. However, it is hard to conclude either MLD or ILD as the main dominator. In particular, both MLD and ILD display an annual cycle while BLT presents a semi-annual cycle in the western sector. In the eastern sector, both MLD and ILD increase from March to August and decrease from September to February, while BLT increases from March to November and decreases from December to February. Thus, the impacts of MLD and ILD on the BLT is dependent on the seasons. On the other hand, from their interannual time series, there is no BLT (negative) in the years with deeper MLD, while prominent BLT exists in the years with deeper ILD. We also calculated the correlation coefficients between BLT and MLD and ILD, which are -0.07 and 0.47 in the west sector and -0.25 and 0.38 in the east sector. The interannual variation of BLT is mainly related to the ILD variation in the TIO. To further study the seasonal and interannual variations of BLT, we choose the variables in MLD, such as SST and SSS, and thermocline (prominent variations in the deeper ocean)." 3. This paper mainly gave the seasonal and inter-annual variation of BLT in the TIO, but the mechanism for these was not explained enough. As the BLT is affected not only by the SST, SSS, thermocline, but also by the wind stress, rainfall or the fresh water input and even the net heat flux input. More work on mechanism analysis is encouraged.

Thank you for your suggestions. We agree that the forcing from the atmosphere would affect the variations of BLT, such as wind stress, net freshwater flux, and net heat flux. But results in this study have shown that the variation of BLT is not closely related with SST. Thus, in the revised manuscript, we have included the net freshwater flux to explain change of SSS and associate the wind stress to the change of thermocline. The related analysis has been added in section 4 (Line 194-208).

"According to the above analysis, we examined the corresponding atmospheric forcing in the western TIO and eastern TIO, respectively. Figure 9 shows the seasonal evolution of the upper-ocean salinity, MLD, ILD, thermocline, freshwater flux (Precipitation

minus Evaporation, P-E) and the zonal component of the wind stress. In the western TIO, freshwater flux freshens the upper-ocean water from October to April, which in turn, thickens the BLT, consisting of the analysis in Figure 7. On the other hand, westerlies lead to Ekman pumping, which in turn, results in the thinner thermocline (green line) to affect the BLT. In the eastern TIO, the seasonal variation of BLT is more complex than that in the western TIO. Firstly, the seasonal evolution of SSS has a semi-annual feature while freshwater flux does not. This may link to the Indonesian throughflow which brings the freshwater from the Pacific Ocean into the eastern TIO (Shinoda et al., 2012). Secondly, the thermocline presents the opposite seasonal cycle comparing with that in the western TIO, although the zonal wind stress displays a similar seasonal variation in both the western and eastern TIO. Last but not the least, we also noticed that the salinity in the deeper ocean varies similar to the thermocline in the eastern TIO, which is different in the western TIO. Thus, the seasonal variation of BLT in the eastern TIO is not mainly driven by freshwater flux and wind-driven upwelling. Felton et al. ( 2014) have suggested that the seasonal BLT variation in the eastern TIO may be related to the sea level and ILD oscillation."

4. The explanation of the impact of ENSO on the BLT variation is simply accorded to the theory of Xie et al. (2002). Did you find the anomalous easterlies? The Walker Circulation is also needed to be verified.

Many thanks for pointing out this. In Figure 13 of the revised manuscript, the lagged correlations propagate westward. We have added the lag correlation of precipitation and zonal wind stress with the Nino3.4 index in Figure 13. In the lag correlation of precipitation, negative correlation coefficients are in the eastern TIO while positive correlation coefficients are in the western TIO, indicating the downwelling branch of Walker Circulation in the eastern TIO and the upwelling branch of Walker Circulation in the western TIO. The anomalies in easterlies are invoked in the eastern TIO during the El Niño developing and mature phases.

5. Can you show the time series of the SSS, BLT and thermocline during the whole period of 1980-2015? Do the IOD or the ENSO event mainly contribute to the interannual variation?

Thank you for this comment. We have added this time series plot as Figure 11in the revised manuscript. The related analysis is stated follows (Line254-260). "The relationship between BLT and El Niño could also be detected in the time series of BLT, SSS and thermocline anomalies averaged over the western TIO during winter and spring from 1980 to 2015 (Figure 12). During winter (Figure 12a), deeper BLT and thermocline could be found in 1983, 1992, 1998, corresponding to the mature phase of El Niño. During spring (Figure 8b), deeper BLT and thermocline could also be observed in these decaying phase of El Niño years, accompanying with fresher water. On the other hand, the effect of IOD on the interannual variability of BLT could be observed in specific years as well, such as 1983,1998 and 2006. ". Minor comments: 1. Line33-34: What variation?

Thank you for pointing out this. We have added the proper word in front of variation.

2. Line 53-54: INTER-TROPICAL CONVERGENCE ZONE (ITCZ)?

Thank you for your correcting. Corrected.

3. Line 124: 2005-2015 => 2005-2015?

Thank you for pointing out this. We have corrected it.

4. Section 3 might be too short. The author could explain more about figure 1 or move this section to the next section.

Thank you for your suggestion. We have added more content in section 3.

5. Line 213: The caption of figure 4: lead-leg?

Many thanks for pointing out this. We have revised it to lead-lag.

6. Line 252: The lines could be plotted above the shaded area in figure 6.

[Figure]

Thanks for pointing out this. We have re-plotted Figure 10 and Figure 11 in revised manuscript.

Please also note the supplement to this comment:
https://www.ocean-sci-discuss.net/os-2019-12/os-2019-12-AC1-supplement.pdf
* * *
[Figure]

[Figure]

**Fig. 1.** The distributions of MLD_D02 and MLD_DR003 and their difference.

[Figure]

**Fig. 2.** Seasonal variation of MLD_D02 and MLD_DR003 in the TIO.

---

## Author Comment (AC2) · 20 Feb 2020

Dear Dr George Nurser , We would like to thank you for your comments. Your valuable comments help us to improve the quality of this manuscript. In revising the paper, we have carefully considered all comments and suggestions. Major comments:

1. I am not entirely happy with the focus on the SODA reanalysis. It seems to give fields substantially different to ARGO, and I am suspicious of the positive BLT over the whole domain given that substantial parts of the domain (esp to the NW) are areas of net FW loss where salinity should increase towards the surface and there should be no BL. I would like to see further validation of the SODA fields before I can accept this analysis.

Also, the figures need to clearly differentiate areas of no BLT from no data; currently both are white, which is confusing.

Thanks for your concerning and comments. We have evaluated the creditability of SODA BLT by comparing the seasonal spatial feature and variabilities with Argo BLT. Results are presented from Figure 1 to Figure 3 in the revised manuscript. For clarity, we have also plotted the zero contour of BLT in Argo and SODA to differentiate areas of no BLT from no data (see Figure 1). The detailed discussion has been as followings (Lines 107-144 in the revised manuscript).

"BLT calculated by SODA version 3 reanalysis data is assessed against Argo float observation from 2005 to 2015. Figure 1 shows the distributions of the climatological BLT in the TIO for different seasons. BLT climatology obtained from SODA presents a thinner bias in the Bay of Bengal in all four seasons comparing to Argo BLT. This weakened BLT is probably because of lacking the runoff data in the Bay of Bengal (Carton et al., 2018; Carton and Giese, 2008, 2006). SODA BLT fails to capture the BLT feature in the western TIO and northwestern Arabian Sea where no BLT is expected (white areas with green line). However, for the interest area of this study, the BLT in SODA shows a coherent spatial pattern with the Argo BLT in the TIO (55°E-100°E, 5°N -12°S). For instance, thicker BLT locates in the eastern TIO while thinner BLT locates in the western TIO. The seasonal evolution of BLT in the eastern TIO obtained from SODA is consistent with that from Argo as well. The area and intensity of BLT in the eastern TIO experience decreasing from boreal winter to spring and increasing in both boreal summer and autumn. To evaluate the seasonal and interannual variabilities of SODA BLT, the region-averaged BLT over two separated boxes ( from 55 °E -80°E and from 85°E to 100°E, respectively) along with the band between 5°N and 12°S is shown in Figure 2 and Figure 3. In Figure 2, both SODA reanalysis and Argo capture the seasonality of BLT, although the details are somewhat different. In the west sector (55°E -80°E), the thickest BLT is in boreal winter while relatively thin BLT is in boreal spring. In contrast, in the eastern sector (85°E-100°E), the relatively thick BLT occurs in boreal

autumn while the thin BLT occurs in boreal spring and summer. Due to the insufficient temperature-salinity observations, we only compare the interannual variability of the SODA BLT with the Argo during 2005-2010. Two curves show good consistency in both west of 80°E and east of 80°E (Figure 3). Respective correlations between SODA and observation for the west of 80°E and east of 80°E are 0.75 and 0.90, which are statistically significant at the 99.9 % confidence level. Thus, comparisons between SODA and Argo BLT show the SODA capability in representing the seasonal and interannual variability of the BLT in the TIO. In the next section, we will only use SODA reanalysis data to investigate the seasonal and interannual variability of BLT in the TIO (55°E-100°E, 5°N -12°S) from 1980 to 2015. The seasonal and interannual variations of MLD and ILD averaged over the west sector (55°E-80°E, 5°N -12°S) and the east sector (85°E-100°E, 5°N -12°S) have also been calculated in Figure 4 to see which is the dominant role for the BLT variability. However, it is hard to define whether MLD or ILD is the main dominator. In particular, both MLD and ILD display an annual cycle while BLT presents a semi-annual cycle in the western sector. In the eastern sector, both MLD and ILD increase from March to August and decrease from September to February, while BLT increases from March to November and decreases from December to February. Thus, the impacts of MLD and ILD on the BLT is dependent on the seasons. On the other hand, from their interannual time series, there is no BLT (negative) in the years with deeper MLD, while prominent BLT exists in the years with deeper ILD. Then, we calculated their correlation coefficients. Respective correlations of MLD and ILD with BLT are -0.07 and 0.47 in the west sector and -0.25 and 0.38 in the east sector. The interannual variation of BLT is mainly related to the ILD variation in the TIO. To further study the seasonal and interannual variations of BLT, we choose the variables in MLD, such as SST and SSS, and thermocline (prominent variations in the deeper ocean)."

2. It is interesting to see the seasonal variation of the BLT, but there is little discussion of the mechanisms driving it e.g. discussion of why does the SSS change should be linked to freshwater budget changes, while discussion of thermocline depth change

should include e.g. details of changes in Ekman pumping. A couple of figures showing typical vertical profiles would also be useful.

Thank you very much for your suggestion. We have added the seasonal evolution of the upper-ocean salinity, freshwater flux and zonal component of wind stress in Figure 8 and analyzed the change of SSS with freshwater flux and the change of thermocline with wind-driven upwelling (Line194-208). "According to the above analysis, we examined the corresponding atmospheric forcing in the western TIO and eastern TIO, respectively. Figure 9 shows the seasonal evolution of the upper-ocean salinity, MLD, ILD, thermocline, freshwater flux (Precipitation minus Evaporation, P-E), and the zonal component of the wind stress. In the western TIO, freshwater flux freshens the upper-ocean water from October to April, which in turn, thickens the BLT, consisting of the analysis in Figure 7. On the other hand, westerlies lead to Ekman pumping, which in turn, results in the thinner thermocline (green line) to affect the BLT. In the eastern TIO, the seasonal variation of BLT is more complex than that in the western TIO. Firstly, the seasonal evolution of SSS has a semi-annual feature, while freshwater flux does not. This may link to the Indonesian throughflow which brings freshwater from the Pacific Ocean into the eastern TIO (Shinoda et al., 2012). Secondly, the thermocline presents the opposite seasonal cycle comparing with that in the western TIO, although the zonal wind stress displays a similar seasonal variation in both the western and eastern TIO. Last but not least, we also noticed that the salinity in the deeper ocean varies similar to the thermocline in the eastern TIO, which is different in the western TIO. Thus, the seasonal variation of BLT in the eastern TIO is not mainly driven by freshwater flux and wind-driven upwelling. Felton et al. ( 2014) have suggested that the seasonal BLT variation in the eastern TIO may be related to the sea level and ILD oscillation."

3. The discussion of the interannual variability is a little sketchy, but is reasonable. In summary I would like substantial revision validating the data and emphasising more the mechanisms

Thank you very much for your comment. We have added the data evaluation and

mechanisms analysis in the summary (Line 284-298).

"The seasonal and interannual variability of BLT in the TIO was investigated mainly by using the SODA version 3 reanalysis dataset from 1980 to 2015. Although SODA differs in representing the no BLT status near the land mass in the western TIO as shown in Argo, the SODA BLT displays the spatial feature in a good agreement with the Argo BLT. Also, the seasonal and interannual variations of BLT in SODA is consistent with that in Argo. Despite the biases in the spatial feature and variabilities of BLT, SODA is deemed to reproduce overall reasonably well the main characteristics of the BLT in the TIO, and thus it has merits for further exploration of the long-term seasonal and interannual variability of the BLT in the TIO. The contributors to the seasonal variability of BLT is different between the eastern and western TIO. In the eastern TIO, BLT is weakly affected by thermocline change, shown as the deeper thermocline leading to the thicker BLT. This positive correlation between BLT and thermocline is prominent in boreal autumn. In the western Indian Ocean, the factors affecting the BLT change with the season. During boreal autumn, SSS overwhelming SST has a remarkably negative impact on the BLT. The saltier water is, the thinner BLT is. Both SSS and thermocline anomalies make contributions to the BLT during boreal winter through the freshwater flux and the winter monsoon wind-driven upwelling. The positive SSS anomalies shoal BLT while the positive thermocline anomalies thicken BLT. During boreal spring, BLT anomalies are mainly driven by SSS. Meanwhile, there is a weak BLT feedback on SSS anomalies, which is intensified in boreal summer."

Minor comments:

1. The English while readable, is still not great, and could do with reading by a native English speaker. E.g. "composting" actually means allowing vegetables to decay! I think you mean "compositing" or "composited"?

Thank you for your comments. We have improved the English writing.

2. p1, l28–36. Simplify to just stating that previous definitions in terms of temperature

difference have been replaced by new definitions in terms of density difference; leave the details (0.2, 0.03 kg/mËĘ3 etc) to section 2.

Thank you for your suggestion. We have deleted them in the introduction and put them in the section 2 (Line78-80).

3. p2, l23–24. Please explain why zonal SSS gradient is important. Is it to do with Ekman drift?

Thank you for your comments and question. We have added the explanation of the importance of SSS gradient and it is driven by the anomalous wind.

4. p3, l1–8. Please define much more carefully what your definitions of MLD are, and actually write them out as equations. Also, you should mention Kara et al. (2000), as the first paper to use a density criterion based on a temperature criterion: Kara, A. B., P. A. Rochford, and H. E. Hurlburt (2000), An optimal definition for ocean mixed layer depth, C2 OSD Interactive comment Printer-friendly version Discussion paper J. Geophys. Res.-Oceans, 105(C7), 16,803–16,821, doi:10.1029/2000JC900072.

Thank you for your suggestion. We have re-written the definitions of MLD in section 2 and added the reference in the introduction.

5.p3, l11 "reduced systematic errors to a level" What level?

Thank you for pointing out this. We have revised this sentence. See below (Line83-85). "SODA version 3 has reduced systematic errors to the level that are adequate for the no-model statistical objective analysis in the upper ocean and also has improved the accuracy of poleward variability in the tropic (Carton et al., 2018)".

6. p3 l30–35. BLT is zero in the western Indian Ocean in ARGO but not in SODA. SODA BLT see generally excessive. Please compare more fully with ARGO.

Thank you for your comments. We have re-plotted Figure 1 to evaluate the SODA BLT in the tropical Indian Ocean. All related statements have been revised accordingly.

7. p4 l 3 and below. "winter' is confusing. Please use "boreal winter" or "northern winter" (at least the first few times).

Thank you for your correcting. We have corrected it throughout the manuscript.

8. p5 l 21–25. Please spent some time describing the IOD. A figure would help.

Thank you for your suggestion. We just used the Dipole Mode Index (DMI) to do the composition of SSS, BLT and thermocline. Thus, we did not describe the IOD in detail. But we have added the importance of IOD and the description of DMI in section 5 before compositing (Line219-224).

9. p6 l 1–10. Explain how developing and decaying El Nino are defined: it seems you are just considering El Nino and La Nina years.

Many thanks for pointing out this. We have explained the three phases of El Niño in section 5 (Line 239-240). Yes, we just considered the extreme ENSO events. During these years, it is easier to find the relationship between ENSO and BLT in the TIO.

10. Fig. 1. Please differentiate between missing data and zero BLT..

Thank you very much for pointing out this. We have re-plotted Figure 1 and use the zero BLT contour (green line) to differentiate between missing data and zero BLT. The white area with green line means there is no BLT.

11. p11, Fig. 6. Are green shaded areas 95% limits for all years? If so please state this clearly.

Many thanks for your comment. We have added the corresponding description of the Monte- Carlo procedure in the section of method (Line95-98).

Please also note the supplement to this comment:
https://www.ocean-sci-discuss.net/os-2019-12/os-2019-12-AC2-supplement.pdf

---

## Referee Report (RR1)

In this round, I do see great improvements in the manuscript. Still, much patience should be paid in the text-editing and discussion. The followings are some comments.

(1) Line45, 'Comparedto' --> 'Compared to'

(2) Line65, delete 'and SST variation'

(3) Line81, 'ta' --> 'the';    delete 'as the depth'

(4) Line130-133, the time span '2005-2010' is mentioned in the text, but it is different with that shown in Figure3. Which time series is used to calculate the correlation coefficients (0.75 and 0.9) in the text?

(5) Line184-186 and Figure8, Ekman pumping is propotional to the Curl of wind stress (not the zonal wind stress), and pumping would result in a shallow thermocline depth.

(6) Line187, 'THE' --> 'the'

(7) Line216-217, In positive IOD events, more precipitation occurred in the WIO should also contribute to the surface freshwater.

(8) Line407, 'Argo (a) and SODA(b)' --> 'Argo (a-d) and SODA(e-h)'

---

## Author Response (AR2)

**Response letter:**

**Xu YUAN, Xiaolong YU and Zhongbo SU**

**Dear Dr. A.J. George Nurser,**

We would like to thank you for the opportunity to revise our manuscript. Your valuable comments are greatly appreciated, which help us to improve the quality of this manuscript. In revising the paper, we have carefully considered all comments and suggestions. The language has been polished and mistakes in writing and gramma have been corrected to make the manuscript more readable. In this letter. Reviewers' comments are in gray shaded. Our responses are provided in blue color.

**Major comments:**

From the comments of Reviewer A and my own re-reading of the MS, I believe that the MS is still in a ragged state, and requires a careful read through and corrections.

We have carefully revised the manuscript. The language has been polished and the writing has been improved. We believe the manuscript is now readable and in an adequate state for consideration.

For instance, in the definition of BLT in line 76, *MLDDReqDT*02 is specified as the depth of the MLD; I take this to be the Kara et al. definition of MLD in terms of the density difference equivalent to a  $\Delta T=0.2^{\circ}$ . However in the text on line 79 it states that the criterion is  $\Delta$ \rho =0.03.

We have corrected the abbreviation of the mixed layer depth as "MLD" in the definition of BLT in Line 80 and in the definition of MLD in Lines 82-84. "MLD is the mixed layer depth defined by oceanic density at which depth the density is 0.03 kg/m3 larger than that of the surface (de Boyer Montégut et al., 2007; Mignot et al., 2007)."

on p6, line 198-9 it is stated that "On the other hand, westerlies lead to Ekman pumping, which in turn, results in the thinner thermocline (green line)" This looks wrong.

The statement has been corrected as following, "In the meantime, westerlies lead to Ekman pumping in the western TIO, resulting in the thicker thermocline depth (green line) from December to April, which in turn, also makes the BL thicker." See also Lines 184-186.

The last para lines 300-309 needs to make clearer which effects are in the western and eastern Indian Ocean.

We have re-written this paragraph as:

"The interannual variability of BLT exerts a seasonal phase locking pattern during the IOD and ENSO years. In the eastern TIO, thicker BL is led by the deeper thermocline due to wind-induced downwelling during the mature phase of the negative IOD events. In contrast, thinner BL is dominated

by the shallower thermocline due to wind-induced upwelling during the developing and mature phases of the positive IOD events. In the western TIO, the thicker BL is only observed during the mature and decaying phases of the positive IOD events, along with deeper thermocline and fresher surface water. The prominent patterns of BLT in the western TIO can only be detected during the El Niño events. According to the theory of Xie et al. (2002), there is warmer water developing in the eastern tropical Pacific Ocean (El Niño), resulting in the anomalous easterlies and invoking the downwelling Rossby wave along the equatorial TIO. Thereby, the thermocline depth has been deepened in the western TIO, resulting in the thicker BL. This thickening BL hampers the upwelling process and helps to sustain warmer SST. During the decaying phase of El Niño events, there is an anomalous ascending branch of the adjusted Walker circulation in the western TIO. As a result, SSS in the western TIO is decreasing due to abundant precipitation. Consequently, fresher surface water contributes to thickening BL, which in turn, sustains the warmer SST in the western TIO."

Please take account of referee 1's comments, and look at the whole MS carefully; It's important that the MS is in an adequate state for the referee to review.

The manuscript has been carefully revised according to referee's comments. We believe it is now in an adequate state for consideration.

**Dear Referee #1,**

We would like to thank you for the opportunity to revise the manuscript. Your valuable comments are greatly appreciated, which help us improve the quality of this manuscript. In revising the paper, we have carefully considered all comments and suggestions. The language has been polished and mistakes in writing and gramma have been corrected to make the manuscript more readable. In this letter, reviewers' comments are in gray shaded, while our responses to the critics of the referees are provided in blue color.

Minor comments:

In the Abstract(Line9), it is mentioned that 'the western TIO (55°E-75°E, 5°N -12°S)', but in the text, '(55°E-80°E, 5°N -12°S)' is mostly used to represent the western TIO. In Line166, 'western TIO (55°E-75°E, 12°S-5°S)' also appears. Please unify the definition of western TIO.

The definition of the western TIO is now unified in the abstract and the text, where the range of longitude is set between  $55^{\circ}E$  and  $75^{\circ}E$ . See Lines 121-122 in the revised manuscript.

Line10, 'SSS' => 'sea surface salinity (SSS)'

Corrected.

Line24, 'Tropic' => 'tropical'

Corrected.

Line44, 'Sea' => 'sea'

Corrected.

Line53, 'Yuhong et al., 2013' => 'Zhang et al., 2013'

The reference is now properly cited. See Line 64.

Line54-57, For positive IOD, how does the upwelling and less rainfall in the southeastern TIO induce a decrease in salinity? Both isothermal layer and mixed layer become shallower, it doesn't mean BLT will be thinner.

The statement has been revised to be logically robust. "During the positive IOD year (e.g., 2006), thinner BL in the southeastern TIO is mainly led by the shallower thermocline induced by the upwelling Kelvin wave in the presence of weakly shoaling MLD."

See also Lines 56-57 in the revised text.

Line59, The abbreviation 'MLD' should be appeared in Line28.

Corrected.

Line78,' Isothermal Layer Depth (ILD) ' => 'isothermal layer depth (ILD)'

Corrected.

Line79, 'smaller' or 'larger' ?

We have corrected it to "larger".

Line82, 'Asia-pacific data-research center' =>'Asia-Pacific Data-Research Center'

Corrected.

Line86, 'mixed layer depth' => 'MLD'

Corrected.

Line88, 'SST' is used before its definition in Line90

Corrected.

I decide to stop here, because I cann't stand a revised version that involves so many mistakes. It is a waste of time to go on the review of the present version. I believe that the authors didn't take the work seriously and suggest them make a self-examination first.

We have carefully revised the manuscript. The language has been polished and the writing has been improved. We believe the manuscript is now more readable and in an adequate state for consideration.

[revised manuscript text omitted]

---

## Author Response (AR3)

**Response letter:**

Xu YUAN, Xiaolong YU and Zhongbo SU

**Dear Dr. A.J. George Nurser ,**

We want to thank you for organizing quality review of our article and we sincerely appreciate the valuable comments. In our response, reviewers' comments are listed in gray shaded and our responses are provided in blue color.

General comments:

I believe it is now almost publishable. I would just like you to make the changes suggested by the reviewer and also the changes to the English indicated in the attached pdf.

We have carefully checked the manuscript and made changes to the English accordingly as indicated in the commented PDF. We also have been made to address the comments of the reviewer.

**Dear Referee #1,**

We sincerely appreciate your careful reading of our revision. We have made corrections to our previous draft, listing the reviewer's comments in gray shaded and our responses in blue color.

General comments:

In this round, I do see great improvements in the manuscript. Still, much patience should be paid in the text-editing and discussion. The followings are some comments.

Many thanks for your valuable suggestions and comments. In our revised manuscript, the typos are carefully addressed. Detailed descriptions are now included for some of the physical mechanisms.

Minor comments:

Line45, 'Comparedto' --> 'Compared to''

Corrected.

Line65, delete 'and SST variation'

Deleted.

Line81, 'ta' --> 'the'; delete 'as the depth'

Corrected.

Line130-133, the time span '2005-2010' is mentioned in the text, but it is different with that shown in Figure3. Which time series is used to calculate the correlation coefficients (0.75 and 0.9) in the text?'

We used the time span from 2005 to 2015 to calculate the correlation coefficient. We have corrected the time span in the revised manuscript.

Line184-186 and Figure8, Ekman pumping is propotional to the Curl of wind stress (not the zonal wind stress), and pumping would result in a shallow thermocline depth.

We have changed the sentence in Lines 184-187 in the revised manuscript.
"In the meantime, a negative wind stress curl mainly dominated by the zonal wind stress, leads to a weakening Ekman pumping in the western TIO. This weakened Ekman pumping inhibits the upwelling from December to April, resulting in the thicker thermocline depth (green line), which in turn, also makes the BL thicker."

Line187, 'THE' --> 'the'.

Corrected.

Line216-217, In positive IOD events, more precipitation occurred in the WIO should also contribute to the surface freshwater.

We have added this in the revised manuscript in Lines 217-220.

"In the western TIO (Figures 9b, 9d, and 9f), the thicker BL prominently occurs only during the mature phase of positive IOD events that are associated with deeper thermocline and fresher surface water. The deeper thermocline is due to wind-induced downwelling and the fresher surface water is attributed to the westward freshwater advection and more precipitation induced by positive IOD events in the western TIO."

Line407, 'Argo (a) and SODA(b)' --> 'Argo (a-d) and SODA(e-h)'

Corrected.